# Aspirin desensitization in NSAID-exacerbated respiratory disease and its outcomes in the clinical course of asthma: A systematic review of the literature and meta-analysis

Isabel Eraso[1,2], Saveria Sangiovanni[3], Eliana I. Morales[2,4], Liliana Fernández-Trujillo[2,5]*

**1** Department of Internal Medicine, Allergology Service, Fundación Valle del Lili, Cali, Colombia, **2** Faculty of Health Sciences, Universidad Icesi, Cali, Colombia, **3** Clinical Research Center, Fundación Valle del Lili, Cali, Colombia, **4** Department of Internal Medicine, Pulmonology Service, Fundación Valle del Lili, Cali, Colombia, **5** Department of Internal Medicine, Pulmonology Service, Interventional Pulmonology, Fundación Valle del Lili, Cali, Colombia

* liliana.fernandez@fvl.org.co, lilianafernandeztrujillo@gmail.com

## Abstract

### Background

Nonsteroidal anti-inflammatory drug-exacerbated respiratory disease (NERD) might benefit from aspirin desensitization (AD) as an alternative treatment to standard care. However, there is conflicting evidence regarding its role in bronchial symptoms and asthma exacerbations.

### Objective

To analyze the clinical effects of AD in terms of lung function, systemic and inhaled steroid use, the frequency of acute asthma exacerbations, and adverse effects in patients with NERD and asthma.

### Methodology

We identified randomized clinical trials (RCTs) from PubMed, EMBASE, SCOPUS, and EBSCO. We also searched the RCT references for additional studies. Studies comparing AD to placebo in patients with a previous history of pulmonary symptoms triggered by ASA or other NSAIDs or with a positive provocation test to ASA were included.

### Primary results

Five studies with 210 participants with NERD were included in this review. The study duration ranged from 3 to 6 months. Overall, the risk of bias across the included RCTs was low. We identified 3 studies evaluating lung function, 2 of which reported a significant improvement in FEV1 in the AD group after 6 months, while the other reported no difference among the treatments. Due to high heterogeneity, we did not pool the results. The remaining primary outcomes were reported only in a single study each, hindering their interpretation. Secondary outcomes revealed reduced symptom and medication scores in patients with AD.

**Data Availability Statement:** All relevant data are within the paper and its Supporting Information files.

**Funding:** We did not receive any funding for this project.

**Competing interests:** The authors declare no conflict of interests.

**Abbreviations:** NSAID, Nonsteroidal anti-inflammatory drugs; NERD, NSAID-exacerbated respiratory disease; AERD, Aspirin exacerbated respiratory disease; TSLP, Thymic stromal lymphopoietin; PGE2, Prostaglandin E2; COX-1, Cyclooxygenase 1; PGD2, Prostaglandin D2; COX-2, Cyclooxygenase 2; 5 LO, 5 lipoxygenases; ASA, Acetylsalicylic acid; GI, Gastrointestinal; FEV1, Forced expiratory volume in one second; AD, Aspirin desensitization; CysLT, Cysteine leukotriene receptors; RCT, Randomized clinical trial; SMD, Standardized mean difference; ITT, Intention to treat; MD, Mean difference.

## Conclusions

Due to the small number of studies included in this systematic review, conclusions should be made with caution. AD shows a trend towards improving lung function (FEV1) following 6 months of treatment, although no conclusions can be made regarding the use of corticosteroids or the frequency of acute exacerbations. AD appears to reduce both symptom and medication scores. Additional RCTs are needed to fully assess the efficacy of AD in reducing bronchial symptoms in patients with NERD.

## Introduction

### Background

**Description of the condition.** Nonsteroidal anti-inflammatory drug (NSAID)-exacerbated respiratory disease (NERD) was first described 50 years ago by Samter and Beers and was previously known as aspirin-induced asthma or aspirin-exacerbated respiratory disease (AERD) [1]. NERD is a chronic eosinophilic inflammation of the respiratory tract accompanied by nasal polyps, chronic rhinosinusitis and/or asthma, in which the symptoms are typically exacerbated by NSAIDs, including aspirin (ASA) [1, 2]. NERD requires follow-up by several specialties, including pulmonology to manage difficult-to-control asthma, allergology for the management of hypersensitivity to NSAIDs, chronic eosinophilic inflammation, and otolaryngology due to the recurrence of nasal polyps and requirement of surgery [1].

The prevalence of NERD varies from 1.8–44%, depending on the study population and the diagnostic criteria used [1]. The Global Allergy and Asthma European Network GA2LEN reported that 1.94% of the population presents dyspnea associated with NSAID consumption, with an increase in the risk of asthma 4 times greater in patients with NERD [4]. The risk of NERD increases in parallel with the severity of respiratory disease, and these patients have higher hospitalization rates due to asthma (NERD 11.8% vs without NERD 2.4%) [4]. In patients with hypersensitivity to NSAIDs confirmed by a provocation test, the prevalence of asthma increases up to 21% [1–4]. In the univariate analysis of GALEN, an increased risk of asthma [OR 5.50 (4.84–6.26)] and chronic rhinosinusitis [OR 4.28 (3.78–4.84)] was reported in this population [4]. Patients with NERD have twice the risk of having uncontrolled asthma, 60% more asthma exacerbations, 80% more emergency consultations, and 40% more hospitalizations. Additionally, they require more asthma medications and have a poorer quality of life than patients without NERD [5–7].

Among the risk factors for developing NERD, a family history of the disease, the presence of nasal polyps associated with chronic rhinosinusitis and/or asthma, and atopy stand out, alongside a slight predisposition of female patients compared to the male population [1, 8–10]. The disease is usually diagnosed in the 3rd - 4th decade of life, and its natural history involves chronic rhinitis as the first manifestation, progressing to chronic rhinosinusitis, nasal polyps, and anosmia. During the latter period, asthma appears to be triggered [8] and often occurs before acquiring hypersensitivity to NSAIDs. However, there are cases in which hypersensitivity to NSAIDs occurs before the onset of chronic airway disease [1]. Despite NSAID avoidance, patients continue to have asthma exacerbations, loss of smell, and the need for multiple sinus surgeries [11].

After the intake of NSAIDs, symptoms appear within 30–180 minutes, the onset and severity of which are associated with the dose administered. Most patients develop symptoms with 60 mg of acetylsalicylic acid (ASA), but this range varies from 10–300 mg [1, 12, 13].

Manifestations are characterized by high levels of respiratory symptoms, such as nasal congestion and rhinorrhea, and may progress to wheezing, coughing, and dyspnea. In patients with uncontrolled asthma, symptoms appear more rapidly and severely and could potentially lead to fatal outcomes [14]. Urticaria and gastrointestinal (GI) symptoms are also common [1, 9]. Less frequently, patients manifest symptoms associated with alcohol consumption, with eosinophilia sometimes observed in the blood work [1].

Clinical history is key to making a diagnosis of NERD. The appearance of respiratory symptoms 1–2 hours after the consumption of NSAIDs, in patients with adult-onset asthma and with a history of repeated nasal polyposis are key to identifying patients with NERD. If the patient does not meet all criteria or there is doubt in the diagnosis, NSAID hypersensitivity must be confirmed using an oral provocation test, in which increasing doses of the drug are administered following established protocols. This should be done in a safe environment with adequate staff and equipment to ensure an appropriate response to any reactions, such as conjunctivitis or rhinitis, low respiratory symptoms, bronchospasm, a decline in pulmonary function (a decrease in forced expiratory volume in one second (FEV1) by more than 15%), laryngospasm, cutaneous manifestations and systemic symptoms [1, 13]. Oral provocation tests are contraindicated in patients with previous anaphylactic reactions associated with NSAIDs or ASA, uncontrolled asthma with FEV1 <70% of the predicted value, history of chronic renal failure or gastrointestinal bleeding, an exacerbation of asthma in the previous month, pregnancy and present management with beta-blockers [1].

The mainstay for management is the avoidance of the causative drug and other strong COX-1 inhibitor molecules, such as piroxicam, indomethacin, sulindac, tolmetin, ibuprofen, naproxen, fenoprofen, oxazoprin, mefenamic acid, flurbiprofen, diflunisal, ketoprofen, diclofenac, ketorolac, etodolac, nabumetone, and acetylsalicylic acid; typically, NERD patients tolerate selective COX-2 inhibitors, such as celecoxib and etoricoxib. It is of the utmost importance that patients are educated about their disease, understand the alternative medications that are safe for them, and avoid alcohol consumption, as it can worsen symptoms [1]. Specifically, in the asthmatic population, treatment is performed according to various guidelines developed by The National Heart, Lung, and Blood Institute (NHLBI), The Global Initiative for Asthma (GINA), and The British Thoracic Society (BTS), among others [15]. In the GALEN cohort study, it was reported that patients with NERD consume more medications for asthma control (26.1 vs 5.6%) [4], and approximately 30% of patients with hypersensitivity of NSAIDs require high doses of inhaled corticoids [1]. Additionally, these patients typically benefit from management with leukotriene antagonists to try to reduce the existing overexpression of cysteinyl leukotrienes, and in particular cases, they require management with biologics, such as omalizumab or anti-IL5 molecules [1, 16]. Concerning the management of chronic rhinosinusitis and nasal polyps, patients with NERD are more resistant to the usual pharmacological treatments, such as intranasal steroids, also requiring oral steroids for control of the disease. These patients frequently require multiple surgical reinterventions for the recurrence of nasal polyposis (from 24–80% of patients) approximately every 3 years [1, 2].

## Description of the intervention

Use of aspirin desensitization (AD) was initiated in 1922 by Widal et al., who also described the oral provocation test to aspirin in patients with NERD. In 1980, Stevenson et al. reported a decrease in nasal symptom frequency, fewer hospitalizations, and reduced use of systemic steroids in patients with NERD following AD [17]. Long-term administration of ASA after AD enables tolerance to the molecule. Different protocols exist for performing desensitization, but all include the administration of ascending doses of ASA at intervals of 90–120 minutes until a

**Table 1. Aspirin desensitization protocol (table extracted from Kowalski ML, Agache I, Bavbek S, et al. Diagnosis and management of NSAID-Exacerbated Respiratory Disease (N-ERD)—a EAACI position paper. Allergy Eur J Allergy Clin Immunol. 2019).**

| Time | Day 1 | Day 2 |
|---|---|---|
| **9:00 AM** | 20–40 mg | 100–160 mg |
| **11:00 AM** | 40–60 mg | 160–325 mg |
| **01:00 PM** | 60–100 mg | 325 mg |

reaction or the target dose is reached within 1–3 days. If a reaction occurs before achieving the target dose, the process must start again the next day (Table 1). During the process, drug-induced reactions become milder and shorter until they disappear [1, 9]. After completing AD, patients must continue to receive a daily dose of ASA, ranging from 300–1300 mg/day, during a prolonged period to avoid loss of tolerance to ASA [1, 2, 13] (Table 1).

While performing AD, ocular, nasal, bronchial, laryngeal, cutaneous and GI symptoms may occur, equal to those triggered by the oral provocation test; therefore, AD should also be performed with caution, following established protocols, in safe environments and by trained personnel [1, 13]. A severe reaction associated with NSAIDs is not considered a contraindication for AD since the severity of previous reactions does not predict future reactions [12]. Additionally, it is recommended that patients be taken for sinus cytoreduction 2 to 4 weeks before AD because desensitization has not been shown to have an impact on polyp size [13].

Given the long-term administration of aspirin, patients may present with 2 common adverse effects during the desensitization process: gastric ulcers secondary to decreased prostaglandin I2 synthesis and inadequate repopulation of gastric mucosal cells in <15% of patients with bleeding, which occurs predominantly in the skin but can also occur in the nose, bronchi or GI tract [9].

## How the intervention might work

In NERD, there is deregulation of the inflammatory and anti-inflammatory mediators produced by the metabolism of arachidonic acid, causing elevated expression of cysteine leukotriene receptors (CysLT), a concomitant increase in mast cells and eosinophils in the tissues and a decrease in the synthesis of PGE2, which functions as an inhibitor of 5-LO and leukotriene production [17–19]. Associated with this is the description that alteration in the inflammatory mediators in NERD is related to overexpression of IL4, which triggers activation of leukotrienes [17].

It has been suggested that AD followed by maintenance of a daily dose of aspirin improves deregulation of arachidonic acid metabolism by reducing activation of tyrosine kinase and generating inhibition of STAT6 phosphorylation, which leads to a decrease in IL4 production with downregulation of CysLT production and reduced expression of the CysLT1 receptor, ultimately leading to attenuation of airway inflammation and clinical improvement [20]. The key players in this process are Th2 lymphocytes, eosinophils, basophils, mastocytes and platelets. In addition, patients with AD followed by daily doses have been shown to exhibit decreased urinary PGD2 levels, which may be related to decreased effector cell chemotaxis within the tissues, since PGD2 is a potent chemotherapeutic factor for TH2 cells and contributes to a large extent to the eosinophilic inflammation observed in patients with NERD [17].

Patients with NERD who benefit from AD as an add-on therapy, followed by the administration of a daily dose of aspirin include patients with moderate-severe asthma, with inadequate control of nasal symptoms, who show little response to pharmacological management, exhibit recurrence of nasal polyps, need systemic corticosteroids for the control of NERD,

require prevention of nasal polyps after surgery or patients who require aspirin for another condition, such as coronary ischemic disease or chronic anti-inflammatory management [1, 2].

Desensitization to aspirin has shown multiple benefits: improved quality of life, reduced symptoms of high congestion, improved sense of smell, decreased polyp formation and need for surgery, decreased use of systemic corticosteroids, and improved asthma control in patients with NERD [2, 13, 21].

Since patients with NERD have difficulties in the management of both asthma and chronic rhinosinusitis, desensitization to ASA is proposed as an option to improve the course of chronic rhinosinusitis and asthma in these patients.

## Why this review is important

Oral provocation testing and AD followed by daily ASA therapy are important tools for both the diagnosis and specific treatment of NERD, which offers clinical benefit to patients [1, 9]. The goals of NERD management are to decrease inflammation in the upper and lower airways, allowing the prevention of nasal polyp formation, secondary sinusitis, and asthma exacerbation. For patients in whom this objective is not achieved with the usual pharmacological treatment or who require continuous oral steroid doses, desensitization is proposed as a therapeutic option for the control of upper and lower respiratory tract symptoms.

Systematic reviews focused on evaluating the efficacy of AD to improve nasosinusal symptoms have been conducted and found to be a valuable adjunct in the management of these patients [22, 23], but there are no reviews focused on evaluating the effect that AD has on the clinical course of asthma with respect to changes in lung function, decreased steroid use and quality of life in asthma patients. Therefore, there is a need to conduct this review based on the studies published to date to clarify the effects of AD in patients with NERD in terms of outcomes in the clinical course of asthma given that this may be an important therapeutic option in these patients [2, 9].

## Objectives

We sought to analyze the clinical effects of AD compared to placebo in terms of lung function, systemic and inhaled steroid use, frequency of acute asthma exacerbations, and adverse effects in patients with NERD and asthma.

## Methods

We developed this systematic review according to a prespecified study protocol, which was only registered at our institution and was not published elsewhere.

## Included studies

For this review, we included published randomized clinical trials (RCTs) with a parallel design. We did not exclude unblinded studies or randomized pilot studies, but we did choose to exclude open-label trials. Studies should have a minimum of 3 months follow-up. We considered manuscripts in both Spanish and English.

## Type of participants

**Inclusion criteria.** We included patients ≥ 18 years old with a diagnosis of asthma associated with chronic rhinosinusitis and nasal polyps, with a previous history of pulmonary symptoms triggered by ASA or other NSAIDs, or with a positive provocation test to ASA.

**Exclusion criteria.**   Patients with a history of GI bleeding, bleeding diathesis, uncontrolled arterial hypertension, chronic renal failure, uncontrolled asthma with FEV1 < 70%, autoimmune disorders, malignancy, or pregnancy were excluded.

Patients with a history of other pulmonary diseases, such as cystic fibrosis or primary ciliary dyskinesia, we also excluded.

## Types of interventions

Patients with NERD were randomized to receive either AD or placebo. Standard treatment consisting of the management of sinonasal symptoms or asthma medications, depending on the needs of each patient, was allowed. Studies with different protocols were included if they used an ascendant aspiring dosage approach for 1 to 3 days, followed by a daily dosage of aspirin ranging from 300 to 1300 mg/day for several months.

## Outcome measures

**Primary outcomes.**   FEV1 during spirometry
Total daily dosage of systemic steroids
Total daily dosage of inhaled steroids
Acute asthma exacerbations
**Secondary outcomes.**   Frequency of symptoms assessed with symptom score
Medication need score
Nonfatal adverse events

## Search methods for identifying studies

**Electronic searches.**   A PubMed, EMBASE, SCOPUS, and EBSCO search was performed in June 2020 using controlled vocabulary (Mesh and Emtree terms), as well as free text with the following search terms: aspirin-exacerbated respiratory disease, aspirin desensitization, acetylsalicylic acid, asthma, and pulmonary disease.

Results were limited to human-based clinical trials, including patients 18 years old or older, written in English or Spanish. References were manually searched for additional relevant studies. We did not filter results by date of publication.

## Data collection and analysis

**Selection of studies.**   Studies were screened by two independent investigators, IE and SS, who analyzed the titles and abstracts for potentially relevant studies. Then, each study was assessed against the preset inclusion and exclusion criteria.

**Data extraction and management.**   Two investigators, IE and SS, extracted the data from all studies using a standardized data collection form. When disagreements were present, a third party was consulted. For each study, the trial design, participant characteristics, type of interventions, and outcomes were assessed. We used Cochrane Review Manager (RevMan 5.1) software to analyze the data [24].

**Assessment of risk of bias in the included studies.**   Bias was assessed according to the recommendations outlined in the Cochrane Handbook for Systematic Reviews of Interventions [25]. We analyzed the following items:
Allocation sequence generation
Concealment of allocation
Blinding of participants and investigators
Incomplete outcome data

Selective outcome reporting

We graded each potential source of bias as low risk, high risk, or unclear risk of bias.

## Measures of treatment effect

**Dichotomous data.** We analyzed dichotomous data variables using Mantel-Haenszel odds ratios using a fixed-effects model with 95% confidence intervals. If substantial heterogeneity was found among the studies, a random effects model was chosen. If count data were not reported as the number of events per participant, we transformed the variable into a continuous variable.

**Continuous data.** Continuous variables were analyzed as fixed effects mean differences with 95% confidence intervals. If substantial heterogeneity was found among the studies, a random effects model was chosen. Two of the analyzed outcomes used different scales, so the standardized mean difference (SMD) was used. Data were collected using intention-to-treat (ITT) analysis when possible.

**Dealing with missing data.** We did not contact the study authors regarding missing data, but we considered this aspect when judging the quality of evidence and in the analysis of results.

**Assessment of heterogeneity.** We evaluated the degree of statistical variation using the $I^2$ statistic.

**Assessment of reporting biases.** For each study, we compared reported outcomes in the methods sections with the published results to check for reporting biases. We did not contact authors for extra information. Furthermore, we considered very few studies to perform funnel plots.

**Data synthesis.** We constructed a table to summarize our primary outcomes, called the "Summary of Findings Table," using GradePro software [26].

## Results

### Description of studies

**Results of the search.** A PubMed, EMBASE, SCOPUS, and EBSCO search was performed in June 2020, which identified a total of 292 articles. After limiting the search using our pre-specified filters, 45 records were found. Two additional records were identified through reference searching. After reviewing the selected studies, 18 records were removed because they were duplicates. Twenty-nine articles were screened in total, 15 of which were fully assessed. Finally, 5 studies were included in the qualitative analysis and 4 in the quantitative analysis [27–31] (see the PRISMA 2009 flow diagram in Fig 1).

**Included studies.** All studies were randomized controlled trials with a parallel design, except for the study by Stevenson et al, which was a crossover trial, which had an appropriate washout period (1 month) to prevent carry-over effects [30]. We decided to exclude such trials from the quantitative analysis due to the trial design and the incompleteness of information on this review´s outcomes. For all studies, there was a combined total of 199 participants. All studies evaluated the efficacy of aspirin desensitization vs placebo in patients with NERD. However, the study by Swierczynska-Krepa et al also performed AD in patients with aspirin-tolerant asthma, which was not taken into account in this systematic review [31]. Therefore, we included a total of 185 participants, of whom 108 patients received AD and 102 received placebo [27–31].

Inclusion criteria were similar across the five studies: participants were ≥ 18 years old with a prior diagnosis of asthma and a history of aspirin hypersensitivity or known aspirin-exacerbated respiratory disease (AERD). To establish aspirin hypersensitivity/AERD, all studies

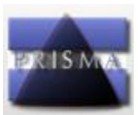

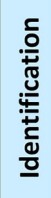

**Identification**

| Records identified through database searching (n = 292 ) | Additional records identified through other sources (n = 2 ) |

Records after duplicates removed (n = 45 )

**Screening**

Records screened (n = 29 )

Records excluded (n = 16 )

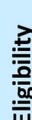

**Eligibility**

Full-text articles assessed for eligibility (n = 27)

Full-text articles excluded (n = 22 )
- Not relevant to article objective (16)
- Review article (5)
- Case report (1)

Studies included in qualitative synthesis (n = 5 )

**Included**

Studies included in quantitative synthesis (meta-analysis) (n = 4 )

*From:* Moher D, Liberati A, Tetzlaff J, Altman DG, The PRISMA Group (2009). *P*referred *R*eporting *I*tems for *S*ystematic Reviews and *M*eta-*A*nalyses: The PRISMA Statement. PLoS Med 6(7): e1000097. doi:10.1371/journal.pmed1000097

**For more information, visit www.prisma-statement.org.**

**Fig 1. PRISMA 2009 flow diagram.**

except the one by Fruth et al used an aspirin challenge test, which was considered positive when clinical symptoms (dyspnea, rhinorrhea, sneezing, ocular secretion, skin flushing, cough, etc.) and a decrease in FEV1 between 15 and 25% from baseline were recorded. In the study by Fruth et al., aspirin intolerance was evaluated in vitro by provoking peripheral leukocytes with the posterior measurement of eicosanoids [28]. Exclusion criteria were described in all but the study by Stevenson et al, reporting consistently that patients with FEV1 values < 70% in spirometry, pregnancy or breastfeeding, history of bleeding diathesis/GI bleeding, malignancy or any chronic diseases of the heart, liver, pancreas, urinary or neurologic system were excluded. Standard asthma medications and therapy for rhinosinusitis symptoms were allowed throughout the studies. Only the study by Swierczynska-Krepa et al explicitly reported that leukotriene modifiers, omalizumab, and immunotherapy were not allowed during the study [31]. Across all studies, participants had a mean age of 38.88 years, and 56.22% were female.

All studies used different doses in the desensitization scheme, although the doses of aspirin were escalated throughout the acute desensitization phase (usually 2 days). Three studies by Mortazavi et al, Esmaeilzadeh et al, and Fruth et al used ascending doses until reaching a pre-specified maximum dose of 120 mg– 180 mg on day 1 and ascending doses until reaching a maximum dose of 475 mg– 800 mg on day 2 [27–29]. The studies by Stevenson et al and Swierc-zynska-Krepa et al used a different approach in which escalation continued until the patient developed a reaction or until the maximum dose was reached (650 mg and 624 mg, respectively) [30, 31]. Maintenance therapy with daily aspirin ranged from 100 mg to 800 mg for 3 to 6 months. For more details, see the characteristics of the included studies table (S1 Table).

## Risk of bias in included studies

For details of the "risk of bias" assessment, refer to the characteristics of the included studies table (S1 Table) and the risk of bias summary (Fig 2).

**Allocation.** In all studies, the randomization method was reported. Regarding allocation concealment, 2 studies by Esmaeilzadeh et al and Swierczynska-Krepa et al were judged to have an unclear risk for selection bias. The randomization list was managed by the unblinded study director in the first study, and no relevant information was reported for the latter [27, 31]. Furthermore, in the studies by Mortazavi et al and Esmaeilzadeh et al, patient baseline characteristics were not reported, which raises some concerns [27, 29]. In the latter study, baseline characteristics referred only to baseline outcome measures, which are not necessarily the most relevant confounding variables [29].

**Blinding.** All studies were considered to have a low risk of performance and detection bias. They were all double-blinded studies in which participants and personnel were not aware of treatment assignment. Placebo was usually made of lactose or starch and looked identical to aspirin. Additionally, the study medication was managed by the hospital's pharmacy. In all included studies, participants and investigators remained blinded until the end of the study.

**Incomplete outcome data.** An attrition rate of < 5% was considered ideal, being acceptable between 5 and 10%. Taking this into consideration, 3 studies were judged as having a low risk for attrition bias. Although the study by Fruth et al had a withdrawal rate of 55% and no sensitivity analysis was performed, all missing data occurred for documented reasons unrelated to the outcome [28]. On the other hand, the studies by Stevenson et al and Swierczynska-Krepa et al reported withdrawal rates of 34% and 17.4%, respectively, and in this case, missingness of data was related to its true value [30, 31].

**Selective reporting.** Complete outcome data were reported according to a prespecified plan in all but one study; only the study by Stevenson et al was judged as having an unclear

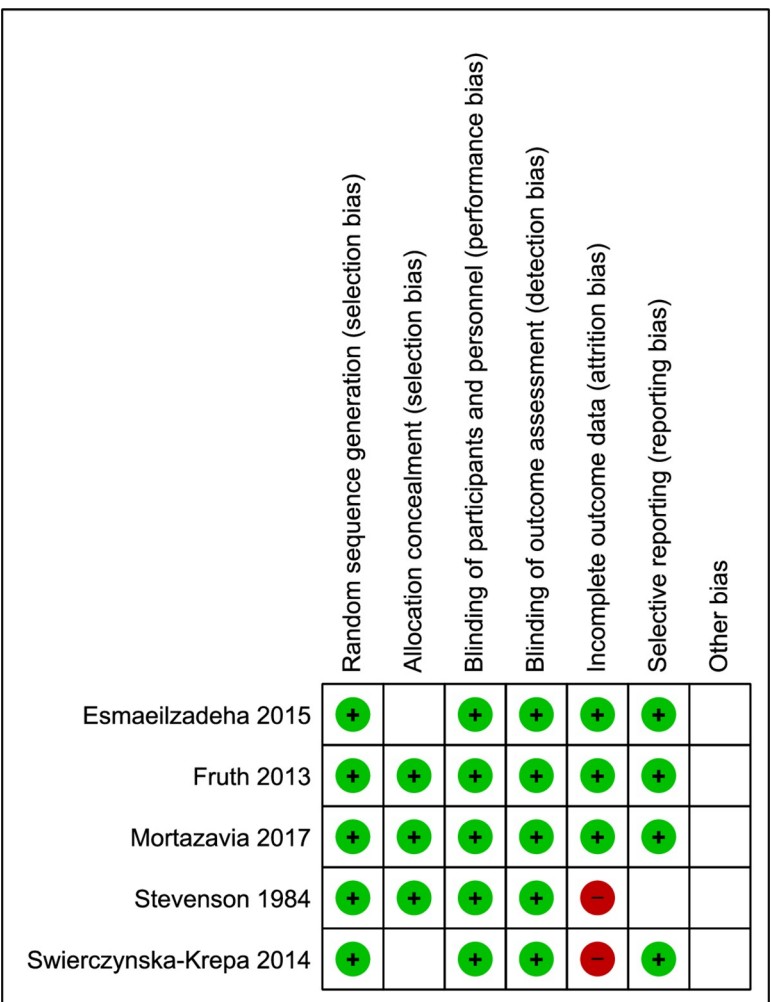

**Fig 2. Risk of bias summary.**

risk of reporting bias since we could not find the study protocol and pulmonary function (specifically VEF1), which was reportedly measured at every follow-up visit but was not depicted in the results section, raising some concerns [30].

### Effects of intervention

The summary of findings (Table 2) shows the main outcomes chosen for this review.

**Primary outcomes.** *Pulmonary function*. Three of the included studies looked at changes in pulmonary functions using VEF1 values (Mortazavi 2017, Esmaelilzadeh 2015, Swierczynska-Krepa, 2014; 86 patients [27, 29, 31]. The study by Swierczynska-Krepa et al did not identify any differences in VEF1 among patients who received AD and those who did not at 6 months [31]. On the other hand, the studies by Mortazavi et al and Esmaelilzadeh et al did report a significant improvement in the experimental group, but in both cases, the increase in FEV1 was seen only after the 6-month follow-up but not earlier [27, 29] (Fig 3). As the study by Swierczynska-Krepa et al reported the results in liters and the other two in % predicted, a standardized mean difference was used. Due to high heterogeneity ($I^2$ = 92%) and the small number of studies included, we felt it was not appropriate to pool the data.

**Table 2. Summary of findings.**

| Summary of findings: |
| --- |

| Aspirin desensitization compared to placebo for Aspirin exacerbated respiratory disease |
| --- |

**Patient or population**: Aspirin exacerbated respiratory disease

**Setting**: Aspirin exacerbated respiratory diseases

**Intervention**: Aspirin desensitization

**Comparison**: placebo

| Outcomes | Anticipated absolute effects* (95% CI) | | Relative effect (95% CI) | № of participants (studies) | Certainty of the evidence (GRADE) | Comments |
| --- | --- | --- | --- | --- | --- | --- |
| | **Risk with placebo** | **Risk with Aspirin desensitization** | | | | |
| Pulmonary function assessed with FEV1 (L or % predicted); mean follow up of 6 months | not pooled | not pooled | - | 86 (3 RCTs) | ⊕⊕◯◯ LOW | High heterogeneity was found among studies, with very few data, being inappropriate to pool the results and draw conclusions. |
| Total daily dosage of inhaled corticosteroids assessed with μg of inhaled budesonide equivalent; follow up of 6 months | The mean total daily dosage of inhaled corticosteroids was **1414 μg** | MD **1039.2 μg lower** (1763.4 lower to 315 lower) | - | 15 (1 RCT) | ⊕⊕◯◯ LOW | A single study reported this outcome, with small sample size, and high attrition rate. The results had wide confidence intervals. |
| Acute asthma exacerbations assessed with frequency of exacerbations; follow up of 6 months | 47 per 100 | **26 per 100** (8 to 58) | **OR 0.40** (0.10 to 1.55) | 38 (1 RCT) | ⊕⊕◯◯ LOW | A single study was found, with small sample size. |

*__The risk in the intervention group__ (and its 95% confidence interval) is based on the assumed risk in the comparison group and the **relative effect** of the intervention (and its 95% CI). CI: Confidence interval; **SMD**: Standardized mean difference; **MD**: Mean difference; **OR**: Odds ratio

GRADE Working Group grades of evidence

**High certainty:** We are very confident that the true effect lies close to that of the estimate of the effect.

**Moderate certainty:** We are moderately confident in the effect estimate: The true effect is likely to be close to the estimate of the effect, but there is a possibility that it is substantially different.

**Low certainty:** Our confidence in the effect estimate is limited: The true effect may be substantially different from the estimate of the effect.

**Very low certainty:** We have very little confidence in the effect estimate: The true effect is likely to be substantially different from the estimate of effect.

*Total daily dose of systemic steroids*. Only the study by Stevenson et al, which was a crossover trial, reported the use of systemic steroids (25 patients) [30]. Although there were available data on the mean daily dosages, it reported how many patients took less, equal, or more prednisone throughout the duration of the study. Of the 25 patients included, 5 were not taking prednisone before the study began. Of those taking prednisone, 5 patients took the same dosage, 9 patients took less, and 6 patients required a higher dose. Of the patients who took a lower dose, an improvement in rhinitis symptoms, asthma symptoms, or both was reported after AD. The remaining patient exhibited a worsening of both symptoms but concomitantly reduced the prednisone dose. Patients who took a higher dose all experienced a worsening of rhinitis and/or

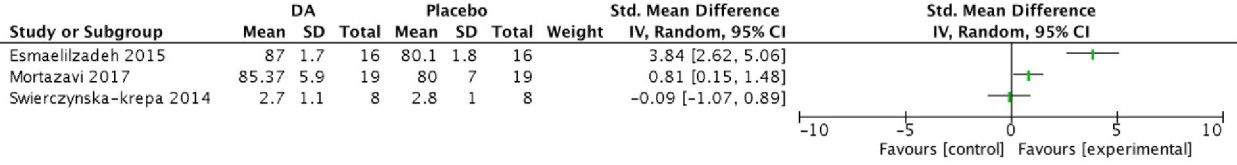

**Fig 3. Forest plot of comparison.** Aspirin desensitization vs placebo, outcome: Pulmonary function (VEF1).

asthmatic symptoms during AD. This study did not report changes in prednisone dosage during the placebo phase, which makes it impossible to further analyze this outcome.

*Total daily dose of inhaled corticosteroids.* Only the study by Swierczynska-Krepa et al reported a daily dosage of inhaled corticosteroids, measured as micrograms (μg) of inhaled budesonide equivalent (15 patients). At 6 months, the daily dosage of inhaled corticosteroids was significantly lower in patients with chronic AD than in the control group, showing a mean difference of 1039.2 μg lower budesonide equivalents, although with a very wide confidence interval (95% CI -1763.4 to -315) (Table 2) [31].

*Acute asthma exacerbations.* Only the study by Mortazavi et al (38 patients) reported the frequency of acute asthma exacerbations throughout the study, which had a follow-up period of 6 months. The study failed to identify a statistically significant difference between the experimental and control groups, with an OR of 0.40 (95% CI 0.10–1.55) favoring the experimental group (Table 2) [29].

**Secondary outcomes.** *Symptom score.* Three studies reported symptom scores for the experimental and control groups [27–29]. The studies by Mortazavi et al and Esmaelilzadeh et al used the same symptom score, which evaluated nasal (itching, sneezing, nasal discharge, nasal obstruction), eye (itching, redness) and bronchial symptoms (cough, wheezing, difficulty breathing) categorized into mild, moderate and severe, with a maximum score of 27 points (a larger score reflected more severe symptoms) [27, 29]. On the other hand, the study by Fruth et al used a questionnaire that evaluated primary and secondary nasal symptoms (nasal obstruction, postnasal drip, cephalgia, impairment of olfactory function) plus paranasal symptoms (coughing and wheezing), with a maximum score of 20 points [28]. For this review, we were primarily interested in bronchial or paranasal symptoms for each symptom score, since our objective was to evaluate the efficacy of AD in asthma outcomes; however, the results were given as an absolute value without discrimination of the source of the symptoms. All studies found the symptom score to be significantly higher in the control group after 6 months of treatment. As the scores were different, we used the standardized mean difference in the quantitative analysis, favoring the experimental group (Fig 4).

*Medication score.* The studies by Mortazavi et al and Esmaelilzadeh et al reported the medication score, in which local and systemic drugs were evaluated [27, 29]. Local medications included eye drops and nasal sprays (1 point for each). Regarding systemic medications, antihistamines, systemic beta-2 agonists, inhaled steroids, and theophylline were included (2 points for each). If the drug was used at its maximum dose, the score for that drug was multiplied by two. That being said, the study by Mortazavi et al found the medication score to be significantly higher in the control group after 6 months of treatment [29]. In the study by Esmaelilzadeh et al, there was a trend towards better medication scores in the experimental group, but it was not significant [27]. In our quantitative analysis, we observed a pooled mean difference of -1.72 (95% CI -2.13 to -1.32), favoring the experimental group (Fig 5).

*Nonfatal adverse events.* All studies reported nonfatal adverse events [27–31], and 12.9% of patients in the experimental group experienced adverse events, with gastrointestinal

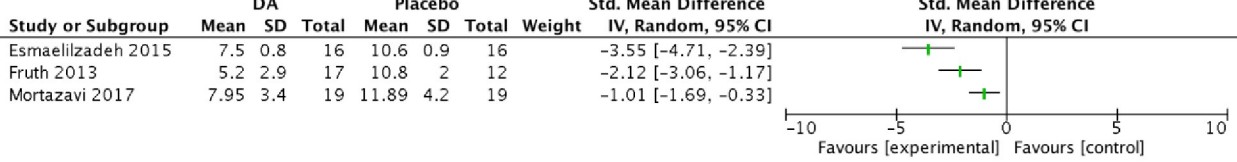

**Fig 4. Forest plot of comparison.** Aspirin desensitization vs placebo, outcome: Symptom score.

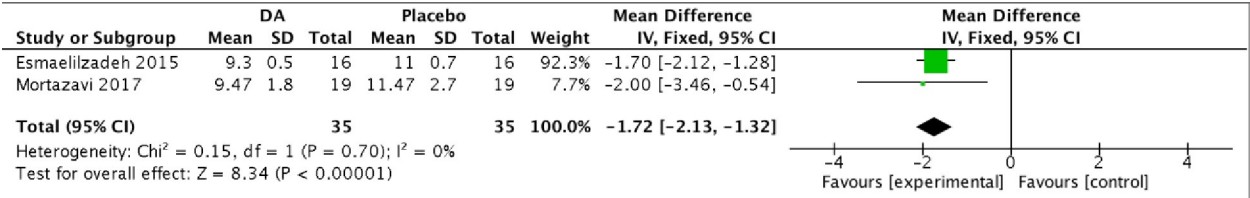

**Fig 5. Forest plot of comparison.** Aspirin desensitization vs placebo, outcome: Medication score.

intolerance being the most common, followed by GI bleeding. The study by Fruth et al reported 0 adverse events in the experimental group [28]. Of the patients in the control group, only 1.9% developed adverse events, corresponding to two patients in the Stevenson et al study. One of those patients presented with uterine bleeding, raising concern of a carryover effect [30]. We included all studies in the quantitative analysis except the one by Stevenson et al, observing a significant difference among groups when the results were pooled, favoring the controls, although with a very wide confidence interval (OR 6.62 (95% CI 1.12–39.25) (Fig 6).

## Discussion

This systematic review suggests that more randomized controlled trials are needed to evaluate the efficacy of AD in patients with NERD in terms of asthma outcomes, particularly lung function, the tapering of systemic and inhaled steroids, and the reduction of acute exacerbations. Five studies have been published to date evaluating the efficacy of AD in patients with NERD in terms of asthma outcomes. Regarding pulmonary function, the results were contradictory since two studies reported an improvement in FEV1 after 6 months of performing AD, while one failed to show a positive change in pulmonary function at the same follow-up time. Nevertheless, AD seems to improve lung function only after a long period of maintenance therapy ($\geq$ 6 months); the studies consistently showed no differences before that time point [27, 29, 31].

Concerning the other primary outcomes, it was not possible to construct forest plots, as we found only one study addressing each outcome. The study by Stevenson et al described a decrease in systemic steroid dose in 9 of 20 patients who were taking steroids at the initiation of the study but unfortunately did not report the average daily dose or the steroid dose during the placebo phase. Additionally, no statistical analysis was performed on this outcome, so it was not possible to draw any conclusions [30]. Regarding the use of inhaled steroids, a reduction in the required dose of budesonide was observed in the AD group, although with low precision, affecting the validity of the results [31]. Furthermore, it appears that the frequency of

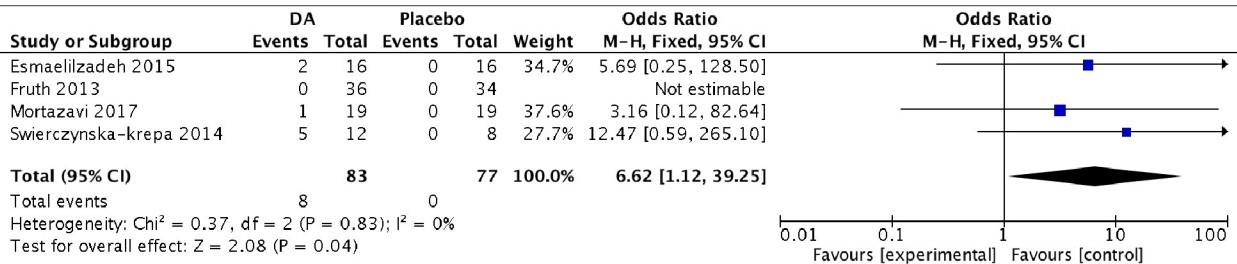

**Fig 6. Forest plot of comparison.** Aspirin desensitization vs placebo, outcome: nonfatal adverse events.

acute asthma exacerbations was not reduced by AD. However, the trial that reported this outcome had a relatively short follow-up period, which could explain the lack of response to treatment [29].

AD appears to reduce symptom scores after 6 months of treatment [27–29]. However, it is impossible to discern whether the reduction was due to an improvement in rhinosinusitis symptoms, asthma symptoms or both.

Regarding adverse events, AD is correlated with GI intolerance and intestinal bleeding [27–29, 31]. However, since some studies reported no adverse events, we consider the possibility that there was underreporting of mild adverse events related to AD.

## Overall completeness and applicability of evidence

Regarding lung function, we found a high degree of statistical heterogeneity, which could not be explained by baseline differences among participants, study duration, or AD protocol, impeding the pooling of data. For outcomes other than lung function, data was scarce. Although it appears that there is no beneficial effect of AD in terms of dosage of systemic and inhaled steroids or frequency of acute exacerbations, we could only find complete information from a single study for each outcome, impeding the purpose of a systematic review. We encourage authors to report on these outcomes, as they have a great impact on the health status and quality of life of the asthmatic population. Even though there was a significant improvement in symptom and medication scores among the participants exposed to AD, they combined both nasal and paranasal/bronchial symptoms. As the majority of evidence from this subject surrounds the efficacy of AD on rhinosinusitis symptoms, it would be of great value to focus future research on bronchial symptoms and medications used for the control of asthma and treatment of acute exacerbations.

## Quality of evidence

This systematic review evaluated data from a total of 210 participants. All studies were included in the qualitative analysis, while four studies were included in the quantitative analysis; the study by Stevenson et al was not included because information on primary outcomes was incomplete [30].

The main issue in the quality of evidence of this systematic review is the high withdrawal rates of patients among individual studies. However, only the studies by Stevenson et al and Swierczynska-Krepa et al were considered to have a high risk of attrition bias because in the other studies, missingness of data was not related to its true value [30, 31]. Another potential source of bias was patient demographic characteristics, which was not depicted in two studies [27, 29]. Since there were no imbalances reported among groups, we did not downgrade the quality of evidence, but we must consider the possibility of chance bias.

## Potential biases in the review process

The strengths of this review lie in the fact that two independent investigators identified, reviewed, and extracted the data. We used multiple databases and manually searched the reference section of the manuscripts to identify other relevant studies. Inclusion and exclusion criteria were prespecified to minimize the risk of not including relevant articles. However, the limitations of this systemic review are that we did not look at unpublished data, and authors of individual studies were not contacted for additional information. Furthermore, as we only wanted to evaluate the efficacy of AD in terms of asthma outcomes, we found very few studies addressing this subject, and within these studies, only 2 had information on the majority of the outcomes. Consequently, there is not enough information to draw any robust conclusions.

## Agreements and disagreements with other studies or reviews

The results of this review are consistent with other studies regarding lung function. In the study by Comert et al, the authors described an increase in FEV1 that was directly proportional to follow-up time (<1-> 3 years) but did not reach statistical significance [2]. Similarly, a systematic review by Chu et al found that AD increased FEV1 compared to placebo, with a mean difference (MD) of 5.78 (95% CI, 2.59 to 8.96), although these findings were also not significant [23].

Although we found only one study reporting the use of systemic steroids after treatment in the experimental group but not the control group, it was impossible to perform a quantitative analysis of this outcome, and the study did report a reduction in the dosage of those who were taking steroids. Similarly, the study by Walters et al reported a statistically significant reduction in systemic steroid use per year, in which 64% of patients who continued AD for a year were able to discontinue systemic steroids [32]. Concerning inhaled steroids, we found one study reporting this outcome, reaching a significant difference, although with a very wide confidence interval [31], consistent with the study by Lee et al, which reported a nonsignificant reduction after 1 year of follow-up with different ASA doses (700 mg or 1300 mg daily) [34].

In our review, a small benefit of AD was observed concerning the decrease in medication consumption, assessed by medication score. The study by Walters et al found a significant decrease in daily medication consumption (SABA, LABA, ICS, ICS/LABA, antihistamines, intranasal steroids, immunotherapy, and systemic steroid) after desensitization [32]. A systematic review by Chu et al was also able to demonstrate a result in favor of AD concerning medication score with an MD 2.95 (1.19–4.71) [23].

AD appears to have a positive effect on asthma control associated with a decrease in symptom score. Likewise, the study by Walters et al reported a significant difference in asthma score, but it had a much longer follow-up period (> 10 years) than the studies included in this review [32]. On the other hand, the study by Lee et al reported an improvement in asthma symptoms after 1 year but did not reach statistical significance [34]. The systematic review by Chu et al showed a low level of certainty regarding asthma control (RR 1.76 [95% CI, 0.51 to 6.06) [23].

Regarding acute asthma exacerbations, we found only one study reporting the frequency of exacerbations, reporting a nonsignificant reduction after AD compared to placebo [29]. Although there is not sufficient information to draw any conclusions, the published literature appears to be inconsistent with our findings. The study by Lee et al reported a significant decrease in asthma hospitalizations per year in patients receiving 700 mg ASA daily or 1300 mg ASA daily [34], similar to the findings by Walters et al, which described a decrease not only in hospitalizations/year but also in emergency department consultations [32]. However, Comert et al, who analyzed 40 patients with NERD who received 300 mg daily of ASA over 3 years, found that AD was useful for the control of upper airway symptoms but found no statistically significant differences in hospitalizations or emergency department visits for asthma symptoms [2].

The results observed in this study are consistent with those reported in the literature, establishing desensitization to ASA as a safe treatment, although with frequent adverse effects, GI manifestations being the most common. A systematic review by Chu et al reported an increased rate of adverse effects in patients who received AD compared to placebo, with gastritis and gastrointestinal bleeding being the main adverse effects [23]. In the study by Walters et al, which had a follow-up period of more than 10 years, 38% of patients who received daily therapy with ASA (325 to 650 mg) discontinued treatment due to adverse effects (primarily peptic ulcer, abdominal discomfort, reflux, and/or minor bleeding), with no mortality [32]. In

the study by Rozsasi et al, none of the patients who received 100 or 300 mg of ASA daily presented adverse effects that caused the suspension of treatment during the first year, but 2 of 39 patients discontinued daily ASA intake due to gastrointestinal adverse effects after the first year [33]. Furthermore, the study by Lee et al compared 700 mg vs 1300 mg daily ASA as maintenance therapy and found that a similar percentage of patients (56% vs 44%) discontinued ASA due to gastrointestinal manifestations, with dyspepsia being the most common cause [34]. Although all studies excluded a clear relationship between long-term AD and episodes of major bleeding at the extracranial and intracranial levels, it should be kept in mind that there is strong evidence confirming that the chronic use of ASA (in the context of primary or secondary prevention of cardiovascular disease) increases the risk of major bleeding at intracranial and extracranial levels [35].

## Conclusions

### Implications for practice

Due to the small number of studies included in this systematic review and the small amount of data we found for the primary outcomes; conclusions should be treated with caution. AD is a treatment option in patients with NERD who do not respond to conventional management and should be individualized to assess the risks and benefits for each patient. AD tends to improve lung function (FEV1) following 6 months of treatment. Regarding the use of systemic and inhaled steroids, we found only one study assessing each outcome, with little to no evidence of the benefit of AD. Acute asthma exacerbations were also analyzed in a single study, with no proven benefit of AD. On the other hand, we did identify a benefit of AD in improving nasal and bronchial symptoms and in reducing total medication need, including a broad spectrum of medications used by this population for upper and lower airway symptoms. Finally, AD is correlated with an increased frequency of adverse effects, primarily of the GI system, consistent with previous evidence regarding the long-term use of ASA. As AD requires long-term maintenance therapy, it is crucial to individualize treatment and consider the patient's comorbidities to prevent and carefully monitor potential adverse events.

### Implications for research

Additional head-to-head studies comparing AD versus standard treatment in patients with NERD are required with respect to lower airway involvement and asthma-related outcomes, since most studies report data evaluating the benefits of AD in rhinosinusitis. Therefore, lung function should always be assessed, as well as the use of systemic and inhaled steroids. Even when studies report daily medication needs (assessed with a medication score that includes a broad range of drug classes), it would be helpful to see where the benefit comes from, if there is any. The same applies to the symptom score, where it should be specified if the benefit is secondary to improvement in nasal or paranasal symptoms. Patient-centered outcomes, such as quality of life, would be of great help in determining the efficacy of AD. Most studies report quality of life using scores such as the sinonasal outcome test (SNOT-20/SNOT-22), which do not apply to asthma-related quality of life. Hence, more relevant instruments should be used in future trials.

If any benefit is seen with AD, it appears to happen after 6 months of treatment; therefore, studies should include longer follow-ups to fully assess the efficacy and safety of the treatment. The latter is of extreme importance, as life-threatening adverse effects will likely occur as maintenance therapy is prolonged. Future studies should aim to determine the minimum effective dose during maintenance therapy to standardize treatment protocols. Additionally, since the clinical response to AD is not the same in all patients, it would be ideal to identify different

phenotypes in patients concerning inflammatory markers to determine the group of patients who would have a better clinical response in terms of lung function and asthma control.

## Supporting information

**S1 Checklist. PRISMA 2009 checklist.**
(DOC)

**S1 Appendix.**
(XLSX)

**S1 File.**
(DOCX)

**S1 Table. Characteristics of included studies.**
(DOCX)

## Author Contributions

**Conceptualization:** Isabel Eraso, Saveria Sangiovanni, Eliana I. Morales, Liliana Fernández-Trujillo.

**Data curation:** Isabel Eraso, Saveria Sangiovanni.

**Formal analysis:** Isabel Eraso, Saveria Sangiovanni.

**Methodology:** Isabel Eraso, Saveria Sangiovanni.

**Supervision:** Eliana I. Morales, Liliana Fernández-Trujillo.

**Writing – original draft:** Isabel Eraso, Saveria Sangiovanni.

**Writing – review & editing:** Isabel Eraso, Saveria Sangiovanni, Eliana I. Morales, Liliana Fernández-Trujillo.

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
