## [Decision Letter · Decision Letter 0]

5 Jan 2021

PONE-D-20-27227

Aspirin desensitization in NSAID-exacerbated respiratory disease and its outcomes in the clinical course of asthma: A systematic review of the literature and meta-analysis.

PLOS ONE

Dear Dr.Fernández-Trujillo ,

Thank you for submitting your manuscript to PLOS ONE. After careful consideration, we feel that it has merit but does not fully meet PLOS ONE’s publication criteria as it currently stands. Therefore, we invite you to submit a revised version of the manuscript that addresses the points raised during the review process and you will find below.

We look forward to receiving your revised manuscript.

Kind regards,

Christophe Leroyer

Academic Editor

PLOS ONE

2. In your methods section, please provide the publication date range that was used to find the manuscripts included in your study.

3. Please confirm that you have included all items recommended in the PRISMA checklist including the full electronic boolean search strategy used to identify studies with all search terms and limits for at least one database. Please attach this as supplementary file.

"None."

"None."

6. We note that you have indicated that data from this study are available upon request. PLOS only allows data to be available upon request if there are legal or ethical restrictions on sharing data publicly. For information on unacceptable data access restrictions, please see http://journals.plos.org/plosone/s/data-availability#loc-unacceptable-data-access-restrictions.

**Comments to the Author**

1. Is the manuscript technically sound, and do the data support the conclusions?

Reviewer #1: Partly

2. Has the statistical analysis been performed appropriately and rigorously? 

Reviewer #1: I Don't Know

3. Have the authors made all data underlying the findings in their manuscript fully available?

Reviewer #1: Yes

4. Is the manuscript presented in an intelligible fashion and written in standard English?

Reviewer #1: No

5. Review Comments to the Author

Reviewer #1: The author performed a systematic review and a meta-analysis about ASA desensitization and the course of asthma in NERD. The authors wanted to assess the impact of ASA desensitization on FEV1, exacerbation rate and inhaled or systemic CSO use.

This is an interesting area though with scare reliable datas. Thus, I wonder wether a meta-analysis is the good choice for assessing the issue.

I have several concerns about this manuscript :

1°) Introduction :

General comment : this section is far too long and should be more synthetic.

line 79 to 86 : I recommend to compare with more recent datas from asthma European cohorts.

line 104 to 114 : references are needed in addition to the one quoted.

line121-122 : skin symptoms is urticaria.

line 126-140 : it should be appropriate to talk about respiratory ASA provocation test. Do they have a place for the diagnosis of NERD ?

line 143 : please mention molecules concerning strong COX-1 inhibitors

line 149 to 151 : please compare with more recent datas from asthma European cohorts.

line 167 : The authors cannot talk about tolerance to ASA since it implies a persistence of the effect after ASA cessation...this is not demonstrated and/or reported to my knowledge.

line 200 to 205 : please precise to the cell type

line 203 : wrong typo "downward" => down-regulation

line 211 to 216 : bette to say that AD is considered as an add-on therapy ?

line 242 to 246 : I'm not sure the authors can say that AD could replace biologics. Furthermore, I fear this assumption is out of the scope of the review.

2°) Method

General comment : this section seems to be rather well-written with regards to my limited knowledge in meta-analysis methodology.

line 335-336 : I do not understand the meaning of that sentence. The author should rephrase it.

3°) Results

General comment : this section is far too long. It should be re-organized in order to highlight important results and give a clear answer to the primary and secondary objectives.

Table 2 should be put in supplemental datas.

line 561 : Watch out typo the lead to misunderstading "but not the one by(...)"

4°) Discussion

General comment : this section should be re-written comparing AD benefits that were demonstrated in upper airways to AD benefits on lower airways.

line 568 to 604 : it sounds like results rather than discussion.

line 606 ti 674 : it sounds like method rather than discussion.

English should be reviewed by an English-native.

6. PLOS authors have the option to publish the peer review history of their article (what does this mean?). If published, this will include your full peer review and any attached files.

Reviewer #1: **Yes: **Luc Colas

---

## [Author Response · Author response to Decision Letter 0]

14 Feb 2021

February 14th, 2020

Editorial Office

PLOS-ONE

Ref: PONE-D-20-27227

Dear Editorial Committee,

Thank you for considering our manuscript for review in PLOS-ONE, and for your comments. They surely helped to improve the quality of our paper. We are answering to the editor’s recommendations. We will approach your suggestions in the same order as you proposed them.

1. Please ensure that your manuscript meets PLOS ONE's style requirements, including those for file naming. The PLOS ONE style templates can be found at…

ANSWER: We organized the manuscript to meet PLOS ONEs requirements. 

2. In your methods section, please provide the publication date range that was used to find the manuscripts included in your study.

ANSWER: We did not filter the results by date of publication. All relevant articles were included. 

3. Please confirm that you have included all items recommended in the PRISMA checklist including the full electronic boolean search strategy used to identify studies with all search terms and limits for at least one database. Please attach this as supplementary file.

ANSWER: In the “Electronic searches” subheading we described the full Boolean search strategy used. Furthermore, we included the PRISMA checklist. 

"None.

ANSWER: We added a financial disclosure within the cover letter, stating we did not receive any funding for this project. 

ANSWER: We added a financial disclosure within the cover letter, stating the authors have declared that no competing interests exist. 

6. We note that you have indicated that data from this study are available upon request.

ANSWER: We will upload the minimal anonymized data set necessary to replicate the study findings as a Supporting Information file. 

Reviewer # 1

7. Is the manuscript presented in an intelligible fashion and written in standard English?

ANSWER: we had a native English speaker revise the manuscript for grammatical and syntax errors. 

8. Introduction: General comment: this section is far too long and should be more synthetic.

ANSWER: We followed the Cochrane handbook for systematic reviews and meta-analysis as a guide to perform this work. Therefore, all included sections of the introduction (Description of the condition, description of the intervention, how the intervention might work, and importance of performing the review) are specified in such handbook and are essential for supporting the importance of the systematic review and defining crucial concepts to construct the methodological aspects. Nevertheless, the introduction was revised and shortened as much as possible. 

9. Introduction: line 79 to 86: I recommend comparing with more recent data from asthma European cohorts.

ANSWER: This paragraph was modified, including data from more recent asthma cohorts in Europe. 

10. Introduction; line 104 to 114: references are needed in addition to the one quoted.

ANSWER: As the reviewer advised that the introduction should be shortened, we decided to eliminate this paragraph as it was not essential, taking into account the objectives of the systematic review. 

11. Introduction; line121-122: skin symptoms is urticaria.

ANSWER: skin manifestations was changed for urticaria. 

12. Introduction; line 126-140: it should be appropriate to talk about respiratory ASA provocation test. Do they have a place for the diagnosis of NERD?

ANSWER: In fact, ASA provocation test is mentioned from line 117-127, as well as their indication in the diagnosis of NERD. 

13. Introduction; line 143: please mention molecules concerning strong COX-1 inhibitors

ANSWER: We added examples of strong COX-Q inhibitors. 

14. Introduction; line 149 to 151: please compare with more recent datas from asthma European cohorts.

ANSWER: We added and compared with data from more recent European cohorts. 

15. Introduction; line 167: The authors cannot talk about tolerance to ASA since it implies a persistence of the effect after ASA cessation...this is not demonstrated and/or reported to my knowledge.

ANSWER: We agree with this comment and modified the manuscript. Tolerance to ASA is dependent on daily usage and is normalized after cessation. 

16. Introduction; line 200 to 205: please precise to the cell type

ANSWER: We added the precise cell types involved in the physiopathology of NERD and ASA desensitization. 

17. Introduction; line 203: wrong typo "downward" => down-regulation

ANSWER: This error was corrected. 

18. Introduction; line 211 to 216: better to say that AD is considered as an add-on therapy?

ANSWER: We agree that AD is an add-on therapy, so we specified this aspect in such paragraph. 

19. Introduction; line 242 to 246: I'm not sure the authors can say that AD could replace biologics. Furthermore, I fear this assumption is out of the scope of the review.

ANSWER: We agree. Although AD is a therapeutic option in a specific type of patients, it does not replace biologics. We clarified this aspect. 

20. Method

General comment: this section seems to be rather well-written with regards to my limited knowledge in meta-analysis methodology.

line 335-336: I do not understand the meaning of that sentence. The author should rephrase it.

ANSWER: We rephrased the sentence and hopefully it is now clear that we are referencing the percentage of missing data in each study and the impact of attrition bias in the study results.

21. Results

General comment: this section is far too long. It should be re-organized in order to highlight important results and give a clear answer to the primary and secondary objectives.

ANSWER: Regarding the length of the results, all included sections were written according to the Cochrane Handbook for systematic reviews and meta-analysis. In systematic reviews, more importance should be given to the methodology of different studies, risk of bias, differences and similarities among studies, in order for the reader to be able to analyze if the results of the included studies are comparable enough to make clinical decisions. Therefore, it would be wrong to focus only on the answers for primary and secondary objectives, since this is not a review of the literature but a systematic review of the quality of evidence. However, we tried to shorten it as much as possible. 

22. Table 2 should be put in supplemental datas.

line 561: Watch out typo the lead to misunderstanding "but not the one by(...)"

ANSWER: Table 2 was taken out of the manuscript and left as table 1 in supplemental data. The typo on line 561 was corrected. 

23. Discussion

General comment: this section should be re-written comparing AD benefits that were demonstrated in upper airways to AD benefits on lower airways. 

ANSWER: Although we tried to reorganize the discussion, we again followed the structured suggested by the Cochrane Handbook for systematic reviews and meta-analysis. Although, we understand the usefulness of comparing AD benefits in the upper airways to AD benefits in the lower airways, this was not the objective of our review. In fact, we only included studies that analyzed AD in terms of asthmatic symptoms. Furthermore, our outcomes were preestablished in the study protocol in terms of lung function, frequency of acute exacerbations, change in inhaled steroids dosing, etc. There is already a systematic review featuring AD in upper airways symptoms, and it could be interested to compare both results in the future. 

line 568 to 604: it sounds like results rather than discussion.

line 606 ti 674: it sounds like method rather than discussion.

ANSWER: we completely agree and reorganized the discussion section, so it sounded less like results or methods. 

Thank you very much for your comments on our manuscript, 

We will attend future recommendations if needed.

Liliana Fernández-Trujillo., MD.

Department of Internal Medicine

Pulmonology Service, Interventional Pulmonology

Fundación Valle del Lili, Cra 98 # 18-49, Tower 2, 4th Floor

Cali, Colombia. Zip code 760032. Phone (+57) 3155006300 

liliana.fernandez@fvl.org.co, lilianafernandeztrujillo@gmail.com

---

## [Editor Report · Decision Letter 1]

16 Feb 2021

Aspirin desensitization in NSAID-exacerbated respiratory disease and its outcomes in the clinical course of asthma: A systematic review of the literature and meta-analysis.

PONE-D-20-27227R1

Dear Dr. Fernández-Trujillo,

We’re pleased to inform you that your manuscript has been judged scientifically suitable for publication and will be formally accepted for publication once it meets all outstanding technical requirements.

Kind regards,

Christophe Leroyer

Academic Editor

PLOS ONE

---

## [Editor Report · Acceptance letter]

22 Feb 2021

PONE-D-20-27227R1 

Aspirin desensitization in NSAID-exacerbated respiratory disease and its outcomes in the clinical course of asthma: A systematic review of the literature and meta-analysis 

Dear Dr. Fernández-Trujillo:

I'm pleased to inform you that your manuscript has been deemed suitable for publication in PLOS ONE. Congratulations! Your manuscript is now with our production department. 

Kind regards, 

on behalf of

Dr. Christophe Leroyer 

Academic Editor

PLOS ONE